# A STRONG DIFFUSION-GENERATED IMAGE DETECTOR VIA CROSS-MODAL REPRESENTATION LEARNING WITH NEIGHBORING PIXEL RELATIONSHIPS

## ABSTRACT

The astonishing proficiency and unprecedented level of realism of diffusion models in creating and manipulating images have undoubtedly drawn concerns. Many methods have been proposed to detect generated images, and most of them take RGB modality as input. Recently, the concept of Neighboring Pixel Relationships (NPR) is proposed to capture and characterize the generalized structural artifacts stemming from up-sampling operations that usually exists in the generation process. The classifier with only the NPR modality as input can achieve good generalizable performance on detecting generated images. Intriguingly, there has been a scarcity of investigative inquiry by considering both RGB and NPR modalities. To this end, this paper leverages features from both RGB and NPR modalities to detect generated images. Specifically, we propose a Strong Diffusion-generated Image Detector (SDID) by taking advantage of two different but complementary representation learning methods, Cross-Modal Contrastive Learning (CMCL) and Cross-Modal Mutual Distillation (CMMD). The CMCL boosts the discrimination of features between real and fake images. While the CMMD simultaneously transfers the learned knowledge between two modalities. CMCL and CMMD work collaboratively so that each modality learns a more comprehensive representation to distinguish real and fake images. Extensive experiments on GenImage, DRCT-2M, and Co-Spy-Bench datasets show that the proposed SDID achieves state-of-the-art results without bells and whistles.

## 1 INTRODUCTION

In recent years, diffusion-based image generation technologies have shown impressive results in generating realistic images. These technologies have provided efficient content editing and generation tools for applications, such as digital creation, commercial advertising, and social entertainment. However, rapid advancements in synthetic technologies have raised significant concerns regarding their potential implications, such as information security and personal privacy. Therefore, there is an urgent need to develop technologies for detecting generated images to maintain a trustworthy cyberspace environment.

Numerous studies have explored various aspects of diffusion models, including controllable generation Zhang et al. (2023), optimizing generative architectures Rombach et al. (2022); Liu et al. (2022b), accelerating the generation process Dhariwal & Nichol (2021); Song et al. (2020), and much more. The wide diversity of image generative models has raised a considerable challenge to the generalizability of detection methods. It demands that generated image detectors should be able to identify images produced by not only the known generative models, but also the newly developed models that have not been involved in the training of detectors. Recent advancements aimed at enhancing this generalization ability include the refinement of detection algorithms Liu et al. (2024a;b); Cheng et al. (2025); Li et al. (2024); Yu et al. (2024a); Nguyen et al. (2024) and the development of pre-trained models Tan et al. (2023); Ojha et al. (2023); Sha et al. (2023); Chen et al. (2024); Yu et al. (2024b); Cui et al. (2025). These methods usually take the RGB modality as input to detect fake images.

Despite these efforts, a conspicuous gap remains in the lack of source-invariant representation exploited from the generator pipeline for generated image detection. Tan et al. (2024b) focuses on achieving source-invariant fake image detection by rethinking artifacts stemming from the up-sampling component of common generation models. They propose a simple but effective artifact representation, termed Neighboring Pixel Relationships (NPR), aimed at achieving generalized fake image detection. NPR serves as the input for training the detection model, and show good general-ization capabilities across different generation models.

In conclusion, both RGB and NPR modalities have advantages in detecting generated images. Intriguingly, there has been a scarcity of inves-tigative inquiry by considering both RGB and NPR modalities to detect generated images. Zhou et al. (2025) is the work using both RGB and NPR modalities. However, Zhou et al. (2025) focuses on leveraging MLLM with RGB and NPR features for the explanations of fake images. To this end, in this paper, as shown in Figure 1, we leverage features from both RGB modality and NPR modality to build a Strong Diffusion-generated Image Detector (SDID) in order to detect generated images. Specifically,

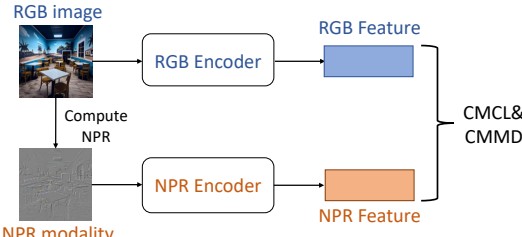

Figure 1: The overview of the proposed SDID with CMCL and CMMD.

we propose two different but complementary representation learning methods, Cross-Modal Con-trastive Learning (CMCL) and Cross-Modal Mutual Distillation (CMMD), to learn a more compre-hensive representation in order to distinguish real and fake images.

Intuitively, taking the RGB feature of a real image as an example, the RGB features and the cor-responding NPR features of the same real image should be closer than the RGB feature of the real image and NPR feature of other fake images. Therefore, we propose CMCL to optimize the RGB embedding space tailored for distinguishing real and generated images. Specifically, as shown in Figure 2 (a), the RGB feature and the NPR feature from the same real image constitute the positive sample pair, while the RGB feature of the real image and the NPR features from other fake images constitute the negative sample pairs. CMCL uses contrastive loss to optimize the positive sample pair and negative sample pairs, in order to boost the discrimination of the RGB feature between real and fake images. Similarly, CMCL can boost the discrimination of the NPR feature between real and fake images.

The above CMCL utilizes the RGB modality and the NPR modality to boost the discrimination of the RGB feature and NPR feature **between** real images and fake images. How-ever, the CMCL can not model the relationships **within** the real images (or fake images). Taking real im-ages as an example, the knowledge in each modality is not been fully explored. Therefore, we propose CMMD to utilize the knowledge in each modality. Specifically, as shown in Figure 2 (b), the neighbor-

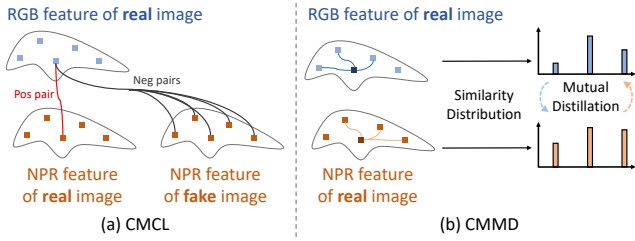

Figure 2: The overview of the proposed CMCL and CMMD with the RGB feature of a real image as an example.

ing similarity distribution is first extracted in each modality. It describes the relationship of the sample embedding with respect to its nearest neighbors in the customized feature space. Based on the relational information, bidirectional knowledge distillation between the two modalities is per-formed via explicit cross-modal consistency constraints.

The CMCL utilizes the cross-modal feature between real images and fake images to boost the dis-crimination of features, while the CMMD utilizes the cross-modal feature within the real images (or fake images) to transfer learned knowledge between RGB and NPR modalities. CMCL and CMMD run synchronously and cooperatively so that each modality learns a more comprehensive representation.

Our contributions are summarized as follows:

1. To fully take advantage of RGB and NPR modalities, we propose Cross-Modal Contrastive Learning (CMCL) to boost the discrimination of the RGB feature and NPR features between real and fake images. (Note that NPR is not claimed as our contribution, since it is proposed in Tan et al. (2024b))

2. To fully utilize the neighboring similarity distribution in each modality of real images (or fake images), we propose Cross-Modal Mutual Distillation (CMMD) to transfer the learned knowledge between RGB and NPR modalities.

3. We propose a Strong Diffusion-generated Image Detector (SDID) by taking advantage of two different but complementary representation learning methods CMCL and CMMD. Extensive experiments on three benchmarks, GenImage, DRCT-2M, and Co-Spy-Bench, verify the effectiveness of the proposed SDID, and show that our method achieves state-of-the-art results without bells and whistles on generated image detection.

## 2 RELATED WORK

### 2.1 AI-GENERATED IMAGE DETECTION

The demand for reliable detection of AI-generated images has existed since the early days of synthetic media. Early approaches primarily relied on spatial-domain artifacts, including color anomalies McCloskey & Albright (2018), saturation inconsistencies McCloskey & Albright (2019), co-occurrence patterns Nataraj et al. (2019), and so on. However, these methods demonstrated limited generalizability as generative models advanced. CNNSpot Wang et al. (2020) made significant progress by showing that a classifier trained only on ProGAN outputs could generalize to unseen GAN architectures when combined with careful preprocessing, postprocessing, and data augmentation. Subsequent work FreqFD Frank et al. (2020) revealed prominent frequency-domain artifacts in GAN-generated images, stemming from architectural upsampling operations.

Recent advances have explored more sophisticated detection paradigms to enhance generalization. DIRE Wang et al. (2023); Ricker et al. (2024); Luo et al. (2024); Cazenavette et al. (2024) proposed a novel approach using differences between images and their reconstructions from a pretrained reconstruction model as discriminative features. PatchCraft Zhong et al. (2023) achieved state-of-the-art generalization by analyzing inter-pixel correlation discrepancies between rich-texture and poor-texture patches. Several studies have successfully leveraged large pretrained models: Uni-vFD Ojha et al. (2023) combined CLIP features with nearest-neighbor classification, FatFormer Liu et al. (2024a) incorporated a forgery-aware adapter into CLIP to integrate local manipulation traces, De-fake Sha et al. (2023) employed CLIP-based multimodal fusion, CLIPMoLE Liu et al. (2024b) adopted a combination of shared and separate LoRAs within an MoE-based structure in deeper ViT blocks, and AIDE Yan et al. (2024) extended this approach by using multiple expert modules to simultaneously capture visual artifacts and noise patterns. Co-Spy Cheng et al. (2025) enhances and integrates existing CLIP-based semantic features and artifact features extracted by VAE. Baraldi et al. (2024) utilizes contrastive learning to train a RGB embedding designed for generated image detection. The above methods usually take the RGB image as input, while Tan et al. (2024b) proposes a simple but effective artifact representation, termed Neighboring Pixel Relationships (NPR), to achieve source-invariant fake image detection by rethinking artifacts stemming from the up-sampling component of common generation models. The detector with only NPR as input shows good generalization capabilities across different generation models. In this paper, we explore leveraging features from both RGB and NPR modality to build a Strong Diffusion-generated Image Detector (SDID) by two representation learning methods.

### 2.2 REPRESENTATION LEARNING

Contrastive learning Chen et al. (2020a); He et al. (2020); Huang et al. (2023) aims to learn feature representation via instance discrimination. It pulls positive pairs closer and pushes negative pairs away. Since no labels are available during self-supervised contrastive learning, two different augmented versions of the same sample are treated as a positive pair, and samples from different instances are considered to be negative. In MoCo He et al. (2020) and MoCo v2 Chen et al. (2020b),

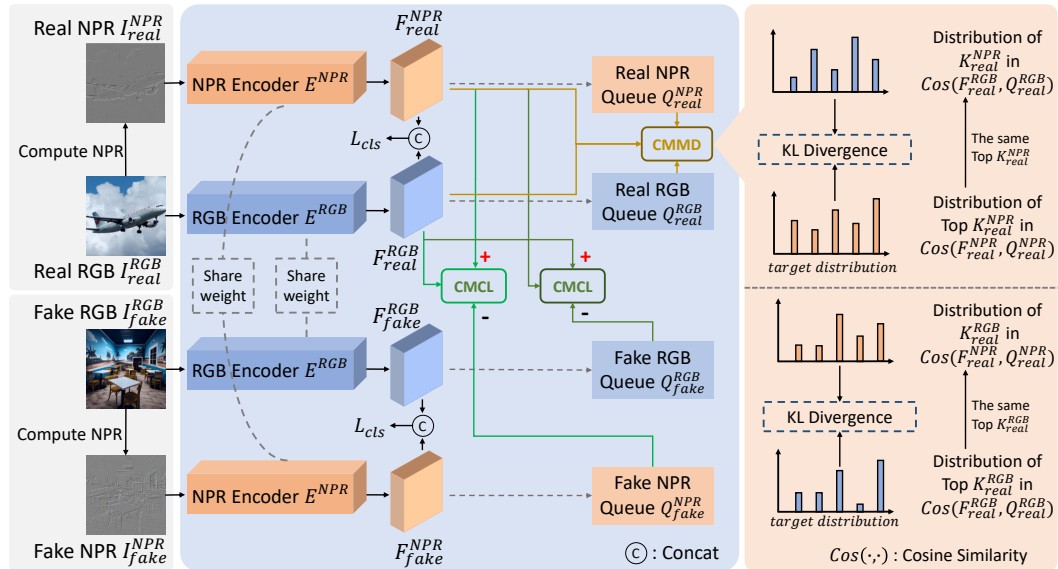

Figure 3: The overall pipeline of SDID. We only use the real images as examples to illustrate the proposed CMCL and CMMD for convenience. The CMCL and CMMD for fake images are similar to those of real images, since the architecture is symmetric.

the negative samples are taken from previous batches and stored in a queue-based memory bank. In contrast, SimCLR Chen et al. (2020a) and MoCo v3 Chen et al. (2021) rely on a larger batch to provide enough negative samples. This paper utilizes the cross-modal (RGB and NPR) features to constitute the positive pair and negative pairs, and explores the cross-modal representation learning to learn more discrimination features between real images and fake images.

## 3 METHOD

### 3.1 FRAMEWORK OVERVIEW

Figure 3 illustrates the overall pipeline of our framework. We only use the real images as examples to illustrate the proposed CMCL and CMMD for convenience. The CMCL and CMMD for fake images are similar to those of real images, since the architecture is symmetric.

Firstly, the NPR modality of real image $I_{real}^{NPR}$ is computed by the RGB modality of real image $I_{real}^{RGB}$. Then, $I_{real}^{RGB}$ and $I_{real}^{NPR}$ are passed through the RGB image encoder $E^{RGB}$ and the NPR image encoder $E^{NPR}$ to extract the RGB feature $F_{real}^{RGB}$ and the NPR feature $F_{real}^{NPR}$, respectively. The $F_{real}^{RGB}$ and $F_{real}^{NPR}$ are stored in queue $Q_{real}^{RGB}$ and $Q_{real}^{NPR}$, respectively. Both $Q_{real}^{RGB}$ and $Q_{real}^{NPR}$ are defined on the fly with the current batch enqueued and the oldest batch dequeued, during training. Similarly, we can get the $I_{fake}^{RGB}, I_{fake}^{NPR}, F_{fake}^{RGB}, F_{fake}^{NPR}, Q_{fake}^{RGB}, Q_{fake}^{NPR}$ belonging to the fake images.

The $F_{real}^{RGB}$, $F_{real}^{NPR}$, and $Q_{fake}^{NPR}$ are used to perform Cross-Modal Contrastive Learning (CMCL). Similarly, $F_{real}^{NPR}$, $F_{real}^{RGB}$, and $Q_{fake}^{RGB}$ are used to perform CMCL. These two CMCL aim to boost the discrimination of $F_{real}^{RGB}$ and $F_{real}^{NPR}$ between real images and fake images. However, the CMCL can only boost the feature discrimination between real and fake images, and cannot model the relationships within the real images. Therefore, we use $F_{real}^{RGB}$, $F_{real}^{NPR}$, $Q_{real}^{RGB}$, and $Q_{real}^{NPR}$ to perform Cross-Modal Mutual Distillation (CMMD) in order to transfers the learned knowledge between RGB and NPR modalities. Finally, we concatenate the feature $F_{real}^{RGB}$ and $F_{real}^{NPR}$, which are already enhanced by the proposed CMCL and CMMD, to predict the label of the real image with cross-entropy loss $L_{cls}$. After training, only the $E^{RGB}$ and $E^{NPR}$ are reserved for generated image detection. The features from $E^{RGB}$ and $E^{NPR}$ are concatenated to predict whether the input image is real or fake.

## 3.2 Neighboring Pixel Relationships

Neighboring Pixel Relationships (NPR) was first proposed in Tan et al. (2024b) to develop a generalizable artifacts representation for generated image detection. Their findings reveal that the up-sampling operator can produce generalized forgery artifacts. Existing works typically consider its influence on the whole image in the frequency domain. In contrast, they explore the trace of the up-sampling layer from the local image pixels, and present a simple but effective artifact representation NPR to achieve generalized generated image detection. Please refer to Tan et al. (2024b) for detailed motivation and theory.

The NPR of an image is computed from its RGB modality. Specifically, the RGB image is divided into non-overlapping patches with patch sizes $l \times l$ ($l = 2$ as suggested by Tan et al. (2024b)). Each pixel in a patch subtracts the value of the top-left pixel per channel in the same patch. We can get the NPR modality of the image after preprocessing all patches. Please refer to Tan et al. (2024b) for more technical details.

## 3.3 Cross-modal Contrastive Learning

Intuitively, taking the RGB feature $F_{real}^{RGB}$ of a real image as an example, the $F_{real}^{RGB}$ and the corresponding NPR feature $F_{real}^{NPR}$ of the same real image should be closer than $F_{real}^{RGB}$ of the real image and the NPR features of other fake images in $Q_{fake}^{NPR}$. Therefore, we propose Cross-Modal Contrastive Learning (CMCL) to boost the discrimination of RGB features between real images and fake images.

We use InfoNCE He et al. (2020); Chen et al. (2020a) loss as contrastive loss. InfoNCE loss seeks to simultaneously pull close positive views from the same sample and push away negative samples. The loss function of InfoNCE loss for the RGB feature $F_{real}^{RGB}$ of the real image is :

$$L_{CMCL}(F_{real}^{RGB}; F_{real}^{NPR}; Q_{fake}^{NPR}) =$$
$$- log \frac{\exp(F_{real}^{RGB} \cdot F_{real}^{NPR}/\tau)}{\exp(F_{real}^{RGB} \cdot F_{real}^{NPR}/\tau) + \sum_{q_i \in Q_{fake}^{NPR}}^{N_Q} \exp(F_{real}^{RGB} \cdot q_i/\tau)} \quad (1)$$

where $log$ denotes the log operation, and $\tau$ is the temperature constant, which is set to 0.07. $N_Q$ is the number of features stored in the queue $Q_{fake}^{NPR}$. The $F_{real}^{RGB} \cdot F_{real}^{NPR}$ of the same real image constitute the positive pair, and the $F_{real}^{RGB} \cdot q_i$ ($q_i \in Q_{fake}^{NPR}$) constitute the negative pairs. Due to the limitation of GPU memory, the $Q_{fake}^{NPR}$ are defined on the fly with the current batch enqueued and the oldest batch dequeued, during training.

The final CMCL loss is defined as:

$$\begin{aligned} L_{CMCL} =& L_{CMCL}(F_{real}^{RGB}; F_{real}^{NPR}; Q_{fake}^{NPR}) \\ &+ L_{CMCL}(F_{real}^{NPR}; F_{real}^{RGB}; Q_{fake}^{RGB}) \\ &+ L_{CMCL}(F_{fake}^{RGB}; F_{fake}^{NPR}; Q_{real}^{NPR}) \\ &+ L_{CMCL}(F_{fake}^{NPR}; F_{fake}^{RGB}; Q_{real}^{RGB}) \end{aligned} \quad (2)$$

With the proposed CMCL, we can boost the discrimination of both RGB and NPR features between real images and fake images.

## 3.4 Cross-modal Mutual Distillation

The proposed CMCL can boost the discrimination of features **between** real images and fake images. However, CMCL can not model the relationships **within** the real images (or the fake images). In other words, taking real images as an example, the knowledge in each modality has not been fully explored. Therefore, we propose Cross-Modal Mutual Distillation (CMMD) to model the learned knowledge and transfer it between modalities. This enables each modality to receive knowledge from other perspectives.

To perform knowledge distillation between modalities, we first need to model the knowledge learned in each modality in a proper way. It needs to take advantage of the existing cross-modal contrastive

learning framework to avoid introducing excessive computational overhead. We utilize the pairwise relationship between samples for modality-specific knowledge modeling. Specifically, given an embedding $z$ and a set of anchors $A = \{a_i\}_{i=1,2,\cdots,K}$, we compute the cosine similarities between them as $cos(z, a_i) = z \cdot a_i, i = 1, 2, \cdots, K$. The queues in contrastive learning store a handful of embeddings of the same modality. We can easily obtain the required anchors without additional model inference. We select the top $K$ nearest neighbors of $z$ as the anchors. The resulting pairwise similarities are further converted into probability distributions with the temperature constant $\tau$:

$$p_i(z, a_i) = \frac{\exp(cos(z, a_i)/\tau)}{\sum_{j=1}^{K} \exp(cos(z, a_j)/\tau)}, i = 1, 2, \cdots, K. \tag{3}$$

The obtained $\boldsymbol{p}(z, A) = \{p_i(z, a_i)\}_{i=1,2,\cdots,K}$ describes the distribution characteristic around the embedding $z$ in the customized feature space of each modality.

Based on the aforementioned probability distributions, an intuitive way to perform knowledge distillation would be to directly establish consistency constraints between RGB and NPR modalities. Different from previous knowledge distillation approaches that transfer the knowledge of a fixed and well-trained teacher model to the student model, in our approach, the knowledge is continuously updated during training, and each modality acts as both student and teacher. Knowledge distillation from modality RGB to modality NPR for real images is performed by minimizing the following KL divergence:

$$
\begin{aligned}
&L_{CMMD}(F_{real}^{RGB}; Q_{real}^{RGB}; F_{real}^{NPR}; Q_{real}^{NPR}) \\
&= \text{KL}\big(\boldsymbol{p}(F_{real}^{RGB}, Q_{real}^{RGB}) \| \boldsymbol{p}(F_{real}^{NPR}, Q_{real}^{NPR})\big) \\
&= \sum_{i=1}^{K} p_i(F_{real}^{RGB}, (q_{real}^{RGB})_i) \cdot \log \frac{p_i(F_{real}^{RGB}, (q_{real}^{RGB})_i)}{p_i(F_{real}^{NPR}, (q_{real}^{NPR})_i)}.
\end{aligned} \tag{4}
$$

where $(q_{real}^{RGB})_i \in Q_{real}^{RGB}$ and $(q_{real}^{NPR})_i \in Q_{real}^{NPR}$. Since the knowledge distillation works bidirectionally, knowledge distillation from modality NPR to modality RGB is defined as $L_{CMMD}(F_{real}^{NPR}; Q_{real}^{NPR}; F_{real}^{RGB}; Q_{real}^{RGB})$.

Similarly, we can get the distillation loss for fake images. Therefore, the overall CMMD loss is defined as:

$$
\begin{aligned}
L_{CMMD} =\,& L_{CMMD}(F_{real}^{RGB}; Q_{real}^{RGB}; F_{real}^{NPR}; Q_{real}^{NPR}) \\
&+ L_{CMMD}(F_{real}^{NPR}; Q_{real}^{NPR}; F_{real}^{RGB}; Q_{real}^{RGB}) \\
&+ L_{CMMD}(F_{fake}^{RGB}; Q_{fake}^{RGB}; F_{fake}^{NPR}; Q_{fake}^{NPR}) \\
&+ L_{CMMD}(F_{fake}^{NPR}; Q_{fake}^{NPR}; F_{fake}^{RGB}; Q_{fake}^{RGB})
\end{aligned} \tag{5}
$$

### 3.5 TRAINING LOSS

The features from $E^{RGB}$ and $E^{NPR}$ are concatenated to predict whether the input image is real or fake. The corresponding classification loss $L_{cls}$ is the cross-entropy loss. The final loss function to train the proposed SDID is the combination of $L_{cls}$, $L_{CMCL}$, and $L_{CMMD}$:

$$L = L_{cls} + \lambda_1 L_{CMCL} + \lambda_2 L_{CMMD} \tag{6}$$

where $\lambda_1$ and the $\lambda_2$ is used to balance the loss among $L_{cls}$, $L_{CMCL}$, and $L_{CMMD}$. After training, only the $E^{RGB}$ and $E^{NPR}$ are reserved for generated image detection.

## 4 EXPERIMENT

### 4.1 DATASETS AND IMPLEMENTATION DETAILS

Experiments are conducted on three benchmarks: GenImage Zhu et al. (2023), DRCT-2M Chen et al. (2024), and Co-Spy-Bench Cheng et al. (2025). The evaluation metric is measured in accuracy, with a decision threshold of 0.5.

The RGB encoder network is the pre-trained Dinov2-ViT-L-14 with LoRA Hu et al. (2022) to fine-tune. The NPR encoder is the ResNet-101 He et al. (2016) pretrained on ImageNet Deng et al. (2009). The length $N_Q$ of each queue used in CMCL is set to 2048, and the Top $K$ nearest neighbors in Equation 3 is set to 128. The loss weight $\lambda_1$ and $\lambda_2$ are experimentally set to 0.1.

| Method | Midjourney | SDv1.4 | SDv1.5 | ADM | GLIDE | Wukong | VQDM | BigGAN | Avg. |
|---|---|---|---|---|---|---|---|---|---|
| CNNSpot | 84.92 | 99.88 | 99.76 | 53.48 | 53.80 | 99.68 | 55.50 | 49.93 | 74.62 |
| F3Net | 77.85 | 98.99 | 99.08 | 51.20 | 54.87 | 97.92 | 58.99 | 49.21 | 73.51 |
| CLIP | 83.30 | 99.97 | 99.89 | 54.55 | 57.37 | 99.52 | 57.90 | 50.00 | 75.31 |
| DeFake | 79.88 | 98.65 | 98.62 | 71.57 | 78.05 | 98.42 | 78.31 | 74.37 | 84.73 |
| ConvB | 83.55 | **99.99** | **99.92** | 51.75 | 56.27 | 99.92 | 58.41 | 50.00 | 74.98 |
| UnivFD | 91.46 | 96.41 | 96.14 | 58.07 | 73.40 | 94.53 | 67.83 | 57.72 | 79.45 |
| DIRE | 50.40 | **99.99** | **99.92** | 52.32 | 67.23 | **99.98** | 50.10 | 49.99 | 71.24 |
| AIDE | 79.38 | 99.74 | 99.76 | 78.54 | 91.82 | 98.65 | 80.26 | 66.89 | 86.88 |
| DRCT | 91.50 | 95.01 | 94.41 | 79.42 | 89.18 | 94.67 | 90.03 | 81.67 | 89.49 |
| VIB-Net | 88.05 | 99.55 | 99.20 | 73.85 | 74.25 | 98.25 | 89.35 | 91.20 | 89.21 |
| Effort | 82.40 | 99.80 | 99.80 | 78.70 | 93.30 | 97.40 | 91.70 | 77.60 | 91.10 |
| **SDID (ours)** | **96.10** | 99.07 | 99.03 | **87.28** | **96.84** | 98.84 | **97.70** | **98.17** | **96.63** |

Table 1: Results on GenImage. All methods were trained on GenImage/SDv1.4 and evaluated on different testing subsets.

| Test Sets | CNNSpot | F3Net | CLIP | GramNet | DeFake | ConvB | UnivFD | DIRE | DRCT | DLFE | **SDID (ours)** |
|---|---|---|---|---|---|---|---|---|---|---|---|
| LDM | 99.87 | 99.85 | 99.00 | 99.40 | 92.10 | **99.97** | 98.30 | 98.19 | 96.74 | 97.13 | 97.57 |
| SDv1.4 | 99.91 | 99.78 | 99.99 | 99.01 | 99.53 | **100.00** | 96.22 | 99.94 | 96.26 | 97.13 | 96.85 |
| SDv1.5 | 99.90 | 99.79 | 99.96 | 98.84 | 99.51 | **99.97** | 96.33 | 99.65 | 96.33 | 97.13 | 96.58 |
| SDv2 | 97.55 | 88.66 | 94.61 | 95.30 | 89.65 | 95.84 | 93.83 | 68.16 | 94.89 | 97.13 | 97.23 |
| SDXL | 66.25 | 55.85 | 62.08 | 62.63 | 64.02 | 64.44 | 91.01 | 53.84 | 96.24 | **97.13** | 93.73 |
| SDXL-Refiner | 86.55 | 87.37 | 91.43 | 80.68 | 69.24 | 82.00 | 93.91 | 71.93 | 93.46 | **97.13** | 94.43 |
| SD-Turbo | 86.15 | 68.29 | 83.57 | 71.19 | 92.00 | 80.82 | 86.38 | 58.87 | 93.43 | **97.13** | 93.18 |
| SDXL-Turbo | 72.42 | 63.66 | 64.40 | 69.32 | 93.93 | 60.75 | 85.92 | 54.35 | 92.94 | **97.13** | 88.80 |
| LCM-SDv1.5 | 98.26 | 97.39 | 98.97 | 93.05 | 99.13 | 99.27 | 90.44 | **99.78** | 91.17 | 97.13 | 96.25 |
| LCM-SDXL | 61.72 | 54.98 | 57.43 | 57.02 | 70.89 | 62.33 | 88.99 | 59.73 | 95.01 | **97.13** | 91.25 |
| SDv1-Ctrl | 97.96 | 97.98 | 99.74 | 89.97 | 58.98 | **99.80** | 90.41 | 99.65 | 95.60 | 97.08 | 96.97 |
| SDv2-Ctrl | 85.89 | 72.39 | 80.69 | 75.55 | 62.34 | 83.40 | 81.06 | 64.20 | 92.68 | **96.81** | 95.65 |
| SDXL-Ctrl | 82.84 | 81.99 | 82.03 | 82.68 | 66.66 | 73.28 | 89.06 | 59.13 | 91.95 | 94.73 | **96.59** |
| SDv1-DR | 60.93 | 65.42 | 65.83 | 51.23 | 50.12 | 61.65 | 51.96 | 51.99 | **94.10** | 92.26 | 86.74 |
| SDv2-DR | 51.41 | 50.74 | 50.67 | 50.01 | 50.16 | 51.79 | 51.03 | 50.04 | 69.55 | 51.79 | **75.89** |
| SDXL-DR | 50.28 | 50.27 | 50.47 | 50.08 | 50.00 | 50.41 | 50.46 | 49.97 | 57.43 | 49.73 | **71.86** |
| Avg. | 81.12 | 77.13 | 80.05 | 76.62 | 75.52 | 79.11 | 83.46 | 71.23 | 90.49 | 90.86 | **91.85** |

Table 2: Results on DRCT-2M. All methods were trained on DRCT-2M/SDv1.4 and evaluated on different testing subsets.

| Test Sets | CNNSpot | FreqFD | Fusing | LNP | UnivFD | DIRE | FreqNet | NPR | DRCT | CO-SPY | **SDID (ours)** |
|---|---|---|---|---|---|---|---|---|---|---|---|
| LDM | 77.56 | 54.17 | 83.03 | 84.87 | 79.07 | 66.25 | 74.92 | 84.34 | 79.25 | **95.04** | 93.00 |
| SD-v1.4 | 89.95 | 62.91 | **99.16** | 95.92 | 80.87 | 83.55 | 69.20 | 90.90 | 81.18 | 91.95 | 93.16 |
| SD-v1.5 | 89.75 | 62.56 | **99.12** | 96.21 | 80.88 | 83.77 | 68.86 | 91.30 | 81.06 | 91.31 | 93.20 |
| SSD-1B | 66.55 | 49.72 | 53.96 | 79.12 | 76.47 | 56.63 | 49.18 | 47.87 | 75.83 | 83.20 | **92.02** |
| tiny-sd | 66.37 | 52.19 | 77.12 | 81.48 | 76.96 | 63.65 | 63.56 | 88.42 | 79.99 | 84.80 | **93.03** |
| SegMoE-SD | 74.41 | 51.74 | 73.58 | 86.62 | 83.07 | 65.98 | 64.40 | 93.79 | 75.12 | 89.49 | **92.86** |
| small-sd | 70.15 | 52.57 | 82.65 | 80.75 | 77.45 | 68.38 | 65.42 | 89.14 | 81.20 | 85.80 | **92.84** |
| SD-2-1 | 68.14 | 49.93 | 59.32 | 57.19 | 81.74 | 65.25 | 51.62 | 51.31 | 76.12 | 88.53 | **93.09** |
| SD-3-medium | 60.68 | 49.98 | 52.47 | 53.69 | 78.42 | 56.75 | 49.64 | 50.00 | 74.95 | 82.91 | **91.42** |
| SDXL-turbo | 88.67 | 61.34 | 59.69 | 83.47 | 84.31 | 72.97 | 66.62 | 83.57 | 80.36 | 95.39 | 93.01 |
| SD-2 | 65.73 | 49.84 | 55.92 | 54.56 | 73.78 | 60.07 | 50.64 | 51.19 | 75.14 | 83.67 | **92.79** |
| SDXL | 61.79 | 49.70 | 51.01 | 80.33 | 63.64 | 51.95 | 48.84 | 47.75 | 75.18 | 74.12 | **91.40** |
| PG-v2.5-1024 | 53.65 | 49.70 | 50.41 | 79.07 | 78.23 | 52.32 | 48.67 | 47.71 | 71.33 | 88.65 | **93.07** |
| PG-v2-1024 | 63.48 | 49.70 | 52.08 | 52.48 | 78.55 | 56.63 | 48.79 | 48.40 | 66.62 | 89.14 | **93.17** |
| PG-v2-512 | 57.94 | 49.87 | 51.58 | 49.40 | 58.90 | 53.63 | 49.09 | 49.09 | 77.61 | 64.86 | **92.38** |
| PG-v2-256 | 63.30 | 50.19 | 51.10 | 54.77 | 62.99 | 60.32 | 49.47 | 49.74 | 73.55 | 72.92 | **84.13** |
| PAXL-2-1024 | 56.29 | 49.76 | 53.60 | 53.41 | 80.08 | 54.77 | 49.40 | 51.16 | 71.80 | **93.94** | 93.12 |
| PAXL-2-512 | 65.44 | 52.20 | 68.77 | 74.25 | 80.32 | 62.18 | 57.23 | 81.25 | 75.53 | **94.96** | 93.11 |
| LCM-sdxl | 81.55 | 52.44 | 70.75 | 85.46 | 78.04 | 68.57 | 62.29 | 50.87 | 81.05 | **96.20** | 93.15 |
| LCM-sdv1-5 | 92.29 | 68.17 | 81.22 | 90.87 | 79.67 | 79.02 | 76.20 | 93.71 | 79.59 | **97.14** | 93.19 |
| FLUX.1-sch | 56.04 | 50.02 | 51.03 | 54.38 | 72.89 | 56.27 | 50.39 | 53.14 | 68.39 | 85.24 | **88.73** |
| FLUX.1-dev | 57.44 | 50.04 | 52.92 | 54.13 | 75.95 | 56.32 | 49.06 | 50.74 | 70.70 | 86.10 | **88.57** |
| Avg. | 69.42 | 53.12 | 65.02 | 71.93 | 76.47 | 63.42 | 57.43 | 65.70 | 75.98 | 87.06 | **92.02** |

Table 3: Results on Co-Spy-Bench. All methods were trained on DRCT-2M/SDv1.4 and evaluated on different testing subsets.

## 4.2 MAIN RESULTS

**GenImage.** We compare the proposed SDID with methods as follows: CNNSpot Wang et al. (2020), F3Net Qian et al. (2020), CLIP Radford et al. (2021), DeFake Sha et al. (2023), ConvB Liu et al. (2022c), UnivFD Ojha et al. (2023), DIRE Wang et al. (2023), AIDE Yan et al. (2024), DRCT Chen et al. (2024), VIB-Net Zhang et al. (2025) and Effort Yan et al. (2025). The results are shown in Table 1, and the proposed SDID outperforms all other methods by at least 5 points.

**DRCT-2M.** We compare the proposed SDID with methods as follows: CNNSpot Wang et al. (2020), F3Net Qian et al. (2020), CLIP Radford et al. (2021), GramNet Liu et al. (2020), DeFake Sha et al. (2023), ConvB Liu et al. (2022c), UnivFD Ojha et al. (2023), DIRE Wang et al. (2023), DRCT Chen et al. (2024), and DLFE Zhong et al. (2025). As shown in Table 2, the proposed SDID outperforms all other methods by at least 1 point.

**Co-Spy-Bench.** We compare the proposed SDID with methods as follows: CNNSpot Wang et al. (2020), FreqFD Frank et al. (2020), Fusing Ju et al. (2022), LNP Liu et al. (2022a), UnivFD Ojha et al. (2023), DIRE Wang et al. (2023), FreqNet Tan et al. (2024a), NPR Tan et al. (2024b), DRCT Chen et al. (2024), CO-SPY Cheng et al. (2025). Table 3 shows that the proposed SDID outperforms all other methods by at least 5 points.

| RGB | NPR | CMCL | CMMD | GenImage |
|:---:|:---:|:---:|:---:|:---:|
| ✓ | | | | 86.60 |
| | ✓ | | | 87.78 |
| ✓ | ✓ | | | 89.07 |
| ✓ | ✓ | ✓ | | 93.59 |
| ✓ | ✓ | ✓ | ✓ | **96.63** |

Table 4: Ablation study of SDID. All models are trained on the GenImage/SDv1.4 dataset, and the averaged accuracy (%) is reported on all test sets of GenImage.

## 4.3 ABLATION STUDY

**SDID.** As shown in Table 4, when concatenating both RGB and NPR features, the detector achieves 89.07% accuracy, which is better than detector that only uses RGB features (86.60%) or NPR features (87.78%). This shows that both RGB and NPR modalities have advantages in generated image detection. Furthermore, CMCL boosts performance from 89.07% to 93.59%, showing the effectiveness of CMCL. Finally, CMMD improves performance from 93.59% to 96.63%, indicating the effectiveness of CMMD.

**Different Inputs.** We also validate the effectiveness of the proposed CMCL and CMMD on different inputs. As shown in Table 5, "RGB & RGB" denotes that we change the NPR input of SDID to another RGB input. "RGB & High Freq" means that we change the NPR input to the high frequency component of the image, since there are evidences that the high frequency of image is useful to detect fake images Luo et al. (2021); Gao et al. (2024); Tan et al. (2024a). Table 5 shows that the proposed CMCL and CMMD can consistently boost the performance of the detector under different inputs, and the NPR modality has advantages in generated image detection.

| Input | CMCL & CMMD | GenImage |
|:---|:---:|:---:|
| RGB & RGB | | 86.89 |
| RGB & RGB | ✓ | **93.06** |
| RGB & High Freq | | 88.01 |
| RGB & High Freq | ✓ | **94.91** |
| RGB & NPR | | 89.07 |
| RGB & NPR | ✓ | **96.63** |

Table 5: Ablation of different inputs on GenImage.

**CMCL and CMMD.** The differences between Table 6 and Table 4 are that only the RGB features are used to predict whether the input image is real or fake. The NPR features are only used to perform CMCL and CMMD. As shown in Table 6, the proposed CMCL boosts the performance from 86.60% to 91.79% by 5.19 points, while the proposed CMMD further boosts the performance from 91.79% to 94.28% by 2.49 points. Note that all experiments in Table 6 only use the RGB encoder to detect generated images without any new model architecture involved. The proposed CMCL and CMMD with NPR modality can boost the performance of vanilla RGB encoder by 7.68 points without increased time cost during testing.

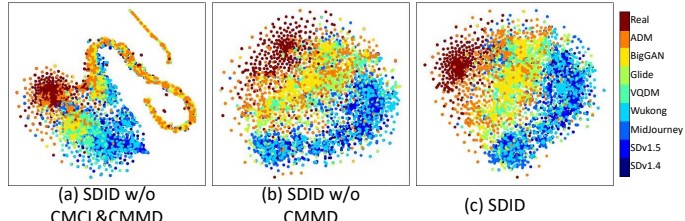

(a) SDID w/o CMCL&CMMD    (b) SDID w/o CMMD    (c) SDID

Figure 4: The t-SNE visualization of embeddings. "SDID w/o CMCL&CMMD" denotes not using CMCL and CMMD. The dark red color denotes the real images.

We also apply t-SNE Van der Maaten & Hinton (2008) to show the embedding distribution among "SDID w/o CMCL&CMMD", "SDID w/o CMMD", and "SDID" on the GenImage benchmark. From Figure 4 (a) and Figure 4 (b), we can see that the embeddings extracted from "SDID w/o CMMD" have better inter-class separability compared with "SDID w/o CMCL&CMMD", further indicating that CMCL can learn more discriminative features. Figure 4 (c) and Figure 4 (b) show that the embeddings of real images extracted from "SDID" have better intra-class compactness compared with "SDID w/o CMMD", further indicating that CMMD can model the relationships within the real images. We also notice that the embeddings between real images (dark red color) and fake

| RGB | CMCL | CMMD | GenImage |
|:---:|:---:|:---:|:---:|
| ✓ | | | 86.60 |
| ✓ | ✓ | | 91.79 |
| ✓ | ✓ | ✓ | **94.28** |

Table 6: Ablation study of CMCL and CMMD. The differences between this Table and Table 4 are that only the RGB features are used to predict whether the input image is real or fake. The NPR features are only used to perform CMCL and CMMD. All models are trained on the GenImage/SDv1.4 dataset, and the averaged accuracy (%) is reported on all test sets of GenImage.

images generated by ADM (orange color) are not perfectly separated in Figure 4 (c), which also aligns with the results in Table 1 where the performance of SDID on ADM test set is the lower than performance of SDID on other 7 test sets of GenImage by at least 9 points.

**Generalized to different RGB encoder.** To validate that the proposed method can be generalized to different RGB encoder, we change the RGB encoder from Dinov2 to the CLIP image encoder in Table 7. The Table 7 shows the proposed method could still boost the performance when using the CLIP image encoder as the RGB encoder. For example, the CMCL boosts the performance from 88.50% to 91.39% by 2.89 points, and the CMMD boosts the performance from 91.39% to 93.54% by 2.15 points.

**Experiments in the APPENDIX A.** (1) More technical details on datasets, data processing, and model training. **(2) Experiments on GAN-generated images.** (3) Experiments on four hyperparameters $N_Q$, Top $K$, $\lambda_1$, and $\lambda_2$.

| RGB | NPR | CMCL | CMMD | GenImage |
|:---:|:---:|:---:|:---:|:---:|
| ✓ | | | | 81.12 |
| | ✓ | | | 87.78 |
| ✓ | ✓ | | | 88.50 |
| ✓ | ✓ | ✓ | | 91.39 |
| ✓ | ✓ | ✓ | ✓ | **93.54** |

Table 7: Generalized to different RGB encoder. The differences between this Table and Table 4 are that the RGB encoder is changed to the CLIP image encoder. All models are trained on the GenImage/SDv1.4 dataset, and the averaged accuracy (%) is reported on all test sets of GenImage.

## 5 CONCLUSION

We propose two representation learning methods to fully take advantage of RGB and NPR modalities to detect generated images. (1) Cross-Modal Contrastive Learning (CMCL) boosts the discrimination of features between real and fake images. (2) Cross-Modal Mutual Distillation transfers the learned knowledge between two modalities. The proposed CMCL and CMMD are complementary and work collaboratively so that each modality learns a more comprehensive representation to distinguish real and fake images. The proposed method is evaluated on three benchmarks with extensive experiments, and achieves state-of-the-art results without bells and whistles on all of them.

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

## A  APPENDIX

### A.1  DATASETS DETAILS

Experiments are conducted on three benchmarks: GenImage Zhu et al. (2023), DRCT-2M Chen et al. (2024), and Co-Spy-Bench Cheng et al. (2025). Drawing from the ImageNet dataset, GenImage utilizes real images and their associated labels to produce various fake images. The training set of GenImage contains fake images generated using Stable Diffusion V1.4. For the test set, we incorporate a variety of generators, including Stable Diffusion V1.4, Stable Diffusion V1.5, GLIDE, VQDM, Wukong, BigGAN, ADM, and Midjourney. DRCT-2M includes 16 types of generated images, with 120k images for each type: 10 types of the currently available SD text-to-image models, 6 types of ControlNet and 3 types diffusion inpainting models. Co-Spy-Bench consists of 22 types text-to-image generative models whose input prompts are derived from the MSCOCO, SBU, Textcups, CC3M and Flickr. The test dataset of Co-Spy-Bench includes 8 types Stable Diffusion variants, 4 types Playground models, 4 types Segmind models, 2 types PixArt models, 2 types LCM models, and 2 types Flux models. Since Co-Spy-Bench hasn't published the official test dataset, we randomly sample 5000 real images from 5 source datasets, and 5000 fake images from every generative model as test dataset.

### A.2  IMPLEMENTATION DETAILS

All detectors take input images of size $308 \times 308$ during training. Images larger than $308 \times 308$ will be center-cropped during testing. Following the post-processing operations of DRCT Chen et al. (2024), a range of data augmentations are conducted during training, including horizontal flipping, Gaussian noise disturbance, Gaussian blurring, random rotation, JPEG compression with random quality, brightness and contrast adjustments, and grid dropout.

During training, we use AdamW optimizer with the learning rate of $3 \times 10^{-4}$. The batch size is set to 32, and the model is trained on one NVIDIA A6000 GPU for only 4 epochs with PyTorch 1.13.0.

| $N_Q$ | GenImage |
|-------|----------|
| 512   | 95.27    |
| 1024  | 96.01    |
| 2048  | **96.63**|
| 4096  | 96.40    |
| 8192  | 96.53    |

((a)) The length $N_Q$ of queue

| Top $K$ | GenImage |
|---------|----------|
| 32      | 95.57    |
| 64      | 96.22    |
| 128     | **96.63**|
| 256     | 96.44    |
| 512     | 95.74    |

((b)) Top $K$ nearest neighbors

Table 8: Ablation study on the length $N_Q$ of each queue in CMCL, and the Top $K$ nearest neighbors in CMMD. All models are trained on the GenImage/SDv1.4 dataset, and the averaged accuracy (%) is reported on all test sets of GenImage.

| $\lambda_1$ | GenImage |
|-------------|----------|
| 1.0         | 93.56    |
| 0.1         | **96.63**|
| 0.01        | 95.87    |
| 0.001       | 93.89    |

((a)) Loss weight $\lambda_1$

| $\lambda_2$ | GenImage |
|-------------|----------|
| 1.0         | 96.11    |
| 0.1         | **96.63**|
| 0.01        | 95.69    |
| 0.001       | 94.34    |

((b)) Loss weight $\lambda_2$

Table 9: Ablation study on the loss weight $\lambda_1$ and $\lambda_2$. All models are trained on the GenImage/SDv1.4 dataset, and the averaged accuracy (%) is reported on all test sets of GenImage.

## A.3 GAN-GENERATED IMAGES

We compare the proposed SDID on ForenSynths dataset Wang et al. (2020) with methods as follows: CNNSpot Wang et al. (2020), FreqFD Frank et al. (2020), PatchFor Chai et al. (2020), F3Net Qian et al. (2020), BiHPF Jeong et al. (2022a), FrePGAN Jeong et al. (2022b), LGrad Tan et al. (2023), UnivFD Ojha et al. (2023), NPR Tan et al. (2024b), FatFormer Liu et al. (2024a), and CoD Jia et al. (2025). As shown in Table 10, the proposed SDID achieves similar results compared with state-of-the-art methods, such as FatFormer Liu et al. (2024a) and CoD Jia et al. (2025). Specifically, the proposed SDID achieves a mean accuracy of 98.3%, demonstrating its effectiveness in GAN-generated image detection.

## A.4 HYPERPARAMETERS

We perform the ablation study of four hyperparameters in the proposed SDID: The length $N_Q$ of each queue in CMCL, the Top $K$ nearest neighbors in CMMD, the loss weight $\lambda_1$ of CMCL, and the loss weight $\lambda_2$ of CMMD. Table 8 shows that $N_Q = 2048$ and Top $K = 128$ achieve the best results, and Table 9 demonstrates that $\lambda_1 = 0.1$ and $\lambda_2 = 0.1$ achieve the best performance.

| Methods | ProGAN | StyleGAN | StyleGAN2 | BigGAN | CycleGAN | StarGAN | GauGAN | Deepfake | Mean |
|---------|--------|----------|-----------|--------|----------|---------|--------|----------|------|
| CNNSpot Wang et al. (2020) | 91.4 | 63.8 | 76.4 | 52.9 | 72.7 | 63.8 | 63.9 | 51.7 | 67.1 |
| FreqFD Frank et al. (2020) | 90.3 | 74.5 | 73.1 | 88.7 | 75.5 | 99.5 | 69.2 | 60.7 | 78.9 |
| PatchFor Chai et al. (2020) | 97.8 | 82.6 | 83.6 | 64.7 | 74.5 | 100.0 | 57.2 | 85.0 | 80.7 |
| F3Net Qian et al. (2020) | 99.4 | 92.6 | 88.0 | 65.3 | 76.4 | 100.0 | 58.1 | 63.5 | 80.4 |
| BiHPF Jeong et al. (2022a) | 90.7 | 76.9 | 76.2 | 84.9 | 81.9 | 94.4 | 69.5 | 54.4 | 78.6 |
| FrePGAN Jeong et al. (2022b) | 99.0 | 80.7 | 84.1 | 69.2 | 71.1 | 99.9 | 60.3 | 70.9 | 79.4 |
| LGrad Tan et al. (2023) | 99.9 | 94.8 | 96.0 | 82.9 | 85.3 | 99.6 | 72.4 | 58.0 | 86.1 |
| UnivFD Ojha et al. (2023) | 99.7 | 89.0 | 83.9 | 90.5 | 87.9 | 91.4 | 89.9 | 80.2 | 89.1 |
| NPR Tan et al. (2024b) | 99.8 | 96.3 | 97.3 | 87.5 | 95.0 | 99.7 | 86.6 | 77.4 | 92.5 |
| FatFormer Liu et al. (2024a) | **99.9** | 97.2 | **98.8** | **99.5** | **99.3** | 99.8 | 99.4 | 93.2 | **98.4** |
| CoD Jia et al. (2025) | **99.9** | 98.2 | 98.6 | 93.6 | 99.1 | **100.0** | **99.9** | 93.4 | 98.2 |
| **SDID (Ours)** | **99.9** | **99.5** | **98.8** | 96.0 | 99.0 | 99.9 | 98.8 | **94.1** | 98.3 |

Table 10: Accuracy comparisons with state-of-the-art methods on ForenSynths dataset. The decision threshold is 0.5.

