# OpenReview forum: "A Strong Diffusion-generated Image Detector via Cross-modal Representation Learning with Neighboring Pixel Relationships"
_ICLR.cc/2026/Conference — ICLR 2026 Conference Withdrawn Submission_

### Official Review · Reviewer_7R39 · 2025-10-29

**Soundness:** 4
**Presentation:** 3
**Contribution:** 3
**Rating:** 6
**Confidence:** 4

**Summary:**

The article proposes a new method for diffusion-generated image detection, which also performs well on GAN-generated images. The contributions revolve around the training objective, which incorporates two novel formulations: Cross-Modal Contrastive Learning, which “boost the discrimination of features between real images and fake images” and Cross-Modal Mutual Distilation, which enables knowledge transfer between modalities by aligning their feature distributions. Both formulations take advantage of NPR-augmented copies of the training samples for real and fake images, improving robustness and feature representation. The method achieves state-of-the-art average accuracy on 3 diffusion-based forged image detection benchmarks.

**Strengths:**

- The method achieves state-of-the-art results on 3 benchmarks, which outperforms the previous performances by 5 percentage points on 2 of those 3 benchmarks.
 - Effective utilization of RGB images and their NPR augmentations in a contrastive manner shows performance improvement over previous state-of-the-art methods.
 - Despite not being mentioned in the main paper, the method achieves near state-of-the-art level performance on GAN-generated images as well (mentioned in the Appendix).
 - The method is straightforward yet not trivial, effectively integrating several valuable ideas from prior works both within and beyond the forgery detection literature.
 - Informative ablation studies are provided to quantify the contribution of components of the method.
 - The paper, for the most part, is easy to follow, and the method is clearly described.

**Weaknesses:**

- CMCL loss has been partially proposed before [1] (eq. 1), if we consider that NPR inputs are augmented versions of the real or fake data samples. However, the CMCL loss includes L_{\text{CMCL}}(F^{\text{NPR}}_{\text{real}}, F^{\text{RGB}}_{\text{real}}, Q^{\text{RGB}}_{\text{fake}}) and L_{\text{CMCL}}(F^{\text{NPR}}_{\text{fake}}, F^{\text{RGB}}_{\text{fake}}, Q^{\text{RGB}}_{\text{real}}), where the negative examples are not augmented, but their impact on performance is not discussed. Furthermore, ablations on the performance gains due to these 2 terms would be helpful to justify their contributions.

- The results on each benchmark are hardly discussed, other than stating that they achieved SOTA (e.g. underwhelming in-distribution performance on GenImage and DRCT-2M, even though the accuracy reaches 99.07\% and 96.85\%, respectively; lots of other methods perform better). A more in-depth interpretation of the results would be helpful.

[1] Baraldi, L., Cocchi, F., Cornia, M., Nicolosi, A., & Cucchiara, R. “Contrasting Deepfakes Diffusion via Contrastive Learning and Global-Local Similarities.” ECCV 2024.

**Questions:**

-  The paper’s captions state competing methods were retrained on the same splits, it would help to include a brief appendix table summarizing baseline training hyperparameters and code refs to ensure reproducibility.
 - How do you keep the size of the queues stable, at N_Q through the training procedure? Is the current sample, for which the loss is computed, included in its corresponding queue?
 - Why the CMMD loss is computed for each category, real and fake, separately? Is there a distinguishable difference in performance by including them separately?
 - Can the authors comment on the notably lower performance of SDID on the ADM subset of the GenImage benchmark? The reported score is nearly 9 points lower than the next lowest SDID result, whereas other methods achieve comparable performance on ADM and at least one additional dataset.

---

### Official Review · Reviewer_Ls84 · 2025-10-31

**Soundness:** 2
**Presentation:** 2
**Contribution:** 2
**Rating:** 2
**Confidence:** 5

**Summary:**

The author tackles the task of fake image detecting using a combination of multiple modalities of an image - the RGB and NPR (neighboring pixel relationships) space. Arguing that previous works have used each of them independently but not together, the authors propose two different objective functions to align feature spaces for these two modalities. The first one makes sure that the NPR and RGB feature of the same image are close and those of different images are far. The second objective ensures that the pairwise distances of images in their NPR and RGB spaces remain proportionate, i.e., the two feature spaces for two modalities behave similarly. Feature from these two modalities are concatenated and fed into a classifier for a binary real/fake prediction. Experiments are done to demonstrate the effectiveness of the proposed method on multiple datasets.

**Strengths:**

- Different modalities of an image (RGB vs NPR) might contain different types of signals for real vs fake detection. So, the idea of utilizing information from multiple modalities to make real/fake prediction is a relatively new idea and makes sense.

- The objective functions designed by the authors are reasonable attempts to align the different features spaces. Their mathematical formulation has been explained clearly. Though, the notation for different parts can be a bit confusing at first (line 199-204).

- The proposed method leads to improvement on 3 different benchmarks.

**Weaknesses:**

- Section 4.2 is supposed to be the main section for discussing the results on the different benchmarks. However, there is absolutely no analysis given for Tables 1-3 showing results on the three benchmarks. Most of the section is just repeatedly mentioning the baselines (e.g., CNNSpot Wang et al. is mentioned 3 times) with just a single line mentioning that the proposed method outperforms the others by x%.

- This pretty much sums up the major issue with the paper. It is difficult for a reader to get something meaningful out of the paper. It pretty much reads like authors trying out some tricks and it somehow ends up improving the performance a little bit on some benchmark. Here are some questions that would have been helpful:
    - What should be the fundamental take away from this paper for a reader?
    - Why do the feature spaces of RGB and NPR need to be trained in this particular way, as opposed to some other way?
    - How can this method help in our understanding of what makes an image fake? Are we able to discover some unknown properties of it through this method?
    - In what way does this method address a key limitation of some of the prior works (beyond just the bare minimum of it combining multiple modalities)?

- Furthermore, while this is not a major weakness, the overall approach is quite cumbersome, with many different components (e.g., feature spaces, loss functions). It is not clear how this approach is "without bells and whistles" (line 31). Did the authors mean to say that their approach is more simple, elegant than the baselines? If so, on what basis are they claiming that?

**Questions:**

Please see the weaknesses.

---

> ### Author Response · Authors · 2025-11-12
> **Why using RGB and NPR, and proposing CMCL and CMMD? Clarification of the movitation and contribution.**
>
> ## Why using RGB and NPR, and proposing CMCL and CMMD? Clarification of the movitation and contribution.
> Typically, fake image detectors take RGB images as input, and use backbones like ResNet, CLIP encoder to extract features for fake image detection. Even though these backbones are capable to detect fake images, they are mainly designed to extract the high-level semantic information, rather than inherently designed for fake images detection. In this paper, we want to optimize the embedding space tailored for detecting fake images, via representation learning.
>
> Previous works has shown that RGB is a good input to detect fake images. We notice that NPR is also capable to capture the intrinsic forgery clues, which means that NPR maybe also a good input to perform repersentation learning that aims at learning the embedding space tailored for detecting fake images.
>
> To this end, we propose CMCL and CMMD (two representation learning methods) together to learn the forgery-aware embedding space, rather than high-level semantic features, and use both NPR and RGB as inputs. The CMCL aims at increasing the inter-class separability (real class vs fake class), while the CMMD focuses on learning tightness features within the intra-class, as evidenced by the TSNE in the Figure 4 in main paper.

---

### Official Review · Reviewer_c1TQ · 2025-10-31

**Soundness:** 2
**Presentation:** 2
**Contribution:** 1
**Rating:** 2
**Confidence:** 5

**Summary:**

This paper proposes a Strong Diffusion-generated Image Detector (SDID) by taking advantage of two different but complementary representation learning methods, Cross-Modal Contrastive Learning (CMCL) and Cross-Modal Mutual Distillation (CMMD) for diffusion-generated image detection. The CMCL and CMMD work collaboratively so that each modality learns a more comprehensive representation to distinguish real and fake images. Experiments on GenImage dataset demonstrate the effectiveness of proposed method.

**Strengths:**

1. This paper aims to solve the diffusion-generated image detection task, which is a critical issue in the related area.
2. The RGB domain and NPR modality should both contain information that could be used for detection.
3. Experiments on GenImage dataset demonstrate the effectiveness of the proposed method.

**Weaknesses:**

1. Why choose RGB and NPR features? The reasons are unclear and unexplained, which is very important for this paper. More justifications are needed.
2. Why is fusing RGB and NPR two modalities helpful for detection? There is no proof on this. Can the authors provide any theoretical or empirical proof on why CMCL and CMMD are helpful for representation learning? What's the difference with separating them?
3. Does it really make sense that aligning or distilling between two different spaces for improving detection? And why just concat the feature after encoder? More explanations are needed.
4. The main baseline of this paper is NPR, which makes this paper more like an incremental work. The authors should justify their novelty and difference with NPR.

**Questions:**

Please refer to the weakness part. Since I have many concerns about the motivation, method, and novelty, I currently lean towards negative ratings. I will change my ratings if the authors could justify them properly.

---

> ### Author Response · Authors · 2025-11-12
> **Why using RGB and NPR, and proposing CMCL and CMMD? Clarification of the movitation and contribution.**
>
> ## Why using RGB and NPR, and proposing CMCL and CMMD? Clarification of the movitation and contribution.
>
> Typically, fake image detectors take RGB images as input, and use backbones like ResNet, CLIP encoder to extract features for fake image detection. Even though these backbones are capable to detect fake images, they are mainly designed to extract the high-level semantic information, rather than inherently designed for fake images detection. In this paper, we want to optimize the embedding space tailored for detecting fake images, via representation learning.
>
> Previous works has shown that RGB is a good input to detect fake images. We notice that NPR is also capable to capture the intrinsic forgery clues, which means that NPR maybe also a good input to perform repersentation learning that aims at learning the embedding space tailored for detecting fake images.
>
> To this end, we propose CMCL and CMMD (two representation learning methods) together to learn the forgery-aware embedding space, rather than high-level semantic features, and use both NPR and RGB as inputs. The CMCL aims at increasing the inter-class separability (real class vs fake class), while the CMMD focuses on learning tightness features within the intra-class, as evidenced by the TSNE in the Figure 4 in main paper.

---

### Official Review · Reviewer_CZTy · 2025-10-31

**Soundness:** 1
**Presentation:** 2
**Contribution:** 1
**Rating:** 2
**Confidence:** 4

**Summary:**

Typically, fake image detectors take inputs as RGB images, however there have been works showing the benefits of other domains, such as Neighboring Pixel Relationships (NPR). However, prior works have not studied the influence of both RGB and NPR features together. The authors hopes to leverage both of these feature types together. In order to do so, the work uses Cross-Modal Contrastive Learning in order to ensure alignment between the real and fake images in RGB and NPR extracted features. At the same time, they also propose Cross-Modal Mutual Distillation in order to ensure that the relationships between various real images are preserved. They attempt to validate these claims by evaluating their model on various benchmarks and outperform certain prominent baselines.

**Strengths:**

1. The method achieves good results on a lot of prominent benchmarks, consisting of modern generative models such as FLUX, SDXL etc.

**Weaknesses:**

1. Line 209-210: This motivates the drawbacks with only using CMCL. The key argument is that CMCL alone cannot preserve the relationships among different real images, therefore the CMMD distillation is necessary. However, the drawbacks of not capturing the relationships among different real images is not clear to me. For instance, two real images could represent the same object (say a cat), however, capturing that information would not actually be helpful for fake image detection, since modern generative models can easily produce the same object. It would greatly help if the authors provide further clarity regarding the motivation here.
2. Related to above, the meaning of relationships among different real images is not clear. Relationships are usually heavily influenced by the task of interest, therefore it would be helpful to the readers if further clarity were provided regarding the same.
3. This point assumes that relationships here are talking about semantic/high-level ones. The CMMD (distillation objective), aims to minimize both the forward and backward KL divergences between the NPR and RGB spaces for both the real and fake image distributions. However, the relationships captured by each encoder is also influenced by the training objective (in this case fake image detection). A neural network trained for fake image detection, is likely not going to focus on broader semantics, therefore, the way in which CMMD addresses this "supposed limitation" is unclear to me.
4. The paper currently lacks a detailed analysis of sensitivity to common post-processing operations (JPEG, upsizing/downsizing, blur, WEBP etc). Given that the authors are training with some of these, I would also encourage them to test their method on unseen perturbations such as WEBP compression/resizing. But I am also interested in seeing the performance on the ones that were used during training. The test should ideally conduct a sweep across different levels of compression/resizing. Some relevant experiments can be found in works [1,2].
5. When comparing on the DRCT-2M benchmark (Tab 2), the DRCT baseline achieves a performance of 90.49 accuracy on average. However, this is the result achieved by the DRCT-UnivFD from the original DRCT paper. The Table 1 also reports results on DRCT-ConvB, which happens to be 96.55, which is almost 5% more than the results that the SDID method is able to achieve. I notice that the authors of DRCT have released the ConvB baseline in their github repo. Is there any particular reason for not including this baseline?

Minor Comments,
1. Line 245: It describes log as log. I feel that this clarification is not actually necessary. If the authors feel this clarification is necessary, it would be better to atleast describe it as the natural logarithm.





References,
1. Cozzolino, D., Poggi, G., Corvi, R., Nießner, M., & Verdoliva, L. (2024). Raising the bar of ai-generated image detection with clip (2023). arXiv preprint arXiv:2312.00195.
2. Rajan, A. S., & Lee, Y. J. (2025). Stay-Positive: A Case for Ignoring Real Image Features in Fake Image Detection. arXiv preprint arXiv:2502.07778.

**Questions:**

1. What is the meaning of "relationships among different real images", this is not clear as of now, and it is crucial to one of the papers major technical contributions, therefore it would be helpful if the authors would provide clarity regarding the same.
2. Refer to weaknesses: But the key argument motivating CMMD is not clear, I request the authors to provide further clarity regarding the same.
3. The experiment in Tab 5 is not well explained. For example, in the setting RGB & RGB, it seems to me like these detectors are being trained with CMCL and CMMD with the RGB & RGB input structure. What does "another RGB input" even mean, wouldnt the image remain the same. Additionally, CMCL and CMMD working on these settings seems to suggest that NPR might not actually be important? Which happens to be a core claim made in this paper.
4. Refer Weaknesses: Please also address the reason for not including the DRCT-ConvB baseline

---

> ### Author Response · Authors · 2025-11-12
> **Why using RGB and NPR, and proposing CMCL and CMMD? Clarification of the movitation and contribution.**
>
> ## Why using RGB and NPR, and proposing CMCL and CMMD? Clarification of the movitation and contribution.
>
> Typically, fake image detectors take RGB images as input, and use backbones like ResNet, CLIP encoder to extract features for fake image detection. Even though these backbones are capable to detect fake images, they are mainly designed to extract the high-level semantic information, rather than inherently designed for fake images detection. In this paper, we want to optimize the embedding space tailored for detecting fake images, via representation learning.
>
> Previous works has shown that RGB is a good input to detect fake images. We notice that NPR is also capable to capture the intrinsic forgery clues, which means that NPR maybe also a good input to perform repersentation learning that aims at learning the embedding space tailored for detecting fake images.
>
> To this end, we propose CMCL and CMMD (two representation learning methods) together to learn the forgery-aware embedding space, rather than high-level semantic features, and use both NPR and RGB as inputs. The CMCL aims at increasing the inter-class separability (real class vs fake class), while the CMMD focuses on learning tightness features within the intra-class, as evidenced by the TSNE in the Figure 4 in main paper.
>
> ## Question about the results of DRCT on Table 2.
>
> The 96.55 result of DRCT from the original DRCT paper is trained on fake images generated by Stable Diffusion v2, while the 90.49 result of DRCT from original DRCT paper is trained on fake images generated by Stable Diffusion v1.4, as shown in the caption of Table 1 of the original DRCT paper. The caption of Table 2 in our paper shows that methods are trained on fake images from Stable Diffusion v1.4. Therefore, we report the 90.49 result for a fair comparison.

---

### Note · Authors · 2025-11-14

**Comment:**

The reviewers had a significant misunderstanding about our paper. We responded as soon as the review comments came out. However, it is a pity that we did not receive any response from the reviewers. Therefore, we decide to withdraw the paper.

**Withdrawal Confirmation:**

I have read and agree with the venue's withdrawal policy on behalf of myself and my co-authors.